## [Transparent Peer Review file · Nature Communications]

Genome-centric metagenomics reveals electroactive syntrophs in a conductive particle-dependent consortium from coastal sediments

Corresponding Author: Professor Amelia-Elena Rotaru

Version 0:

Reviewer comments:

Reviewer #1

(Remarks to the Author)

Jovicic and Anestis et al report on the metagenomic characterization of a methanogenic enrichment culture from the Baltic Sea. Prior work has demonstrated the importance of granular activated carbon for high methane production rates in these incubations that arises from a syntrophic relationship between one or two main syntrophic acetate oxidizing bacteria and a single electron accepting methanogen. With the addition of the metagenomic information provided here, Jovicic and Anestis et al provide a hypothesis for how this symbiosis is carried out and compare it with cultivated relatives of the symbionts.

This manuscript adds detail to an interesting system, the methods are sound, and it is good to have genomic follow-up to prior work on these organisms (i.e. ref 15 and 46). There are a few issues that should be addressed before the work is suitable for publication, outlined below.

Major comment:

A major promise made up front in this manuscript is that "This first genomic blueprint of a natural CIET-SAO consortium identifies potential genomic markers (distinct PCCs, MmcA) for in-situ detection.." (Abstract) and later: "The molecular signatures identified here (novel PCC-clusters and MmcA variants) provide some of the first genomic markers for targeted detection." (Implications).

It would be fantastic to have some sort of a diagnostic gene(s) to positively identify the CIET process in genomes/metagenomes. Unfortunately, that claim is delivered on in this manuscript. For a gene to be a diagnostic marker that can be used for targeted detection it needs to either A) be completely unique to that metabolism with no homologs in other organisms, or B) if there are homologs/paralogs outside of organisms that carry out that metabolism, you need to have clearly defined some specific phylogenetic subgroup(s) that correspond to your process of interest. Then, if a novel sequence is found to fall within those groups, you can propose that they too might serve that function.

A) is clearly not the case, both PCCs and MmcA are used by organisms that do not rely on CIET. Yet the authors have not at all attempted to define the groups for option B), so this work in its present form has not delivered on this major claim. While Figures 4 and 5 have phylogenetic trees with colored subclades, this does not help. For MmcA the Baltic one is very closely related to those in methanosarcina that grow in pure culture using a variety of methanogenic pathways, so there's nothing about MmcA that can be called diagnostic for CIET.

For the omcB, it is claimed to be quite different than ones in *G. sulfurreducens* in terms of amino acid identity, but that is fairly standard for multiheme cytochromes, they evolve incredibly fast. Are we to take the omcB subgroup that is colored blue in Fig 4 as the "CIET group"? Do we know that the *Oryzomonas* sp. therein use them for CIET specifically? Doubtful, and the authors have not made that claim.

Nothing presented actually delivers on a useful test for CIET in a genome, so unless the authors come up with a clear explanation supporting this claim it needs to be removed from the manuscript. The methanosarcina looks just like a normal methanosarcina in every way described in the manuscript, and the *Geosyntrophus* maybe has fewer PCCs than some

Geobacter species, but there is no way to look at a genome, see that it has a single PCC, and tell what it is specific for.

Minor comments:

- 1) Line 169-170 "neither of the constituents showed significant sequence identity to any canonical Geobacter." What does "significant" mean, how are they on a tree with Geobacter versions in Fig 4b if there is no significant sequence similarity?
- 2) Line 189-190 "indicating independent evolution of an identical solution." It is very unclear what is being proposed here from an evolutionary standpoint. Is the claim that the gene organization shown in Fig 4a is a completely novel invention of the PCC complex? This seems like a very strange conclusion with such clear gene synteny and homology. This is almost certainly a mixture of rapid evolution of paralogs and/or HGT not a completely de novo "independent" evolution.
- 3) Line 346-347 "PCC clusters that are unrelated to those of canonical Geobacter pointing at a convergent evolution of EET machineries". Sorry to keep harping on this, but it is extremely odd to talk about things being "unrelated" when you have pulled them out with Blast, aligned them to one another and put them on a phylogenetic tree together. This is the rapid evolution of cytochrome c paralogs which is just how these things go, no clear explanation here of how this is "convergent".
- 4) Line 243: "we recovered the MAG of this methanogen..." already been discussing this MAG for a few paragraphs, introducing it here doesn't make sense.
- 5) Line 204: "disposed of the resulting electrons through a streamline, particle-obligate EET-conduit." Has a control been done to show that the Geosyntrophus cannot put electrons onto iron oxides or other electron acceptors? The authors cite prior work that shows the methanosarcina cannot grow directly on acetate even though it has the machinery for it, so there is some experimental support for the claim that the methanosarcina is specialized for CIET. But it seems like a similar experiment would need to be done, but for the bacteria and alternative electron acceptors, before you can claim that Geosyntrophus is specialized for conductive particles over other electron acceptors.
- 6) If you want the Rnf->HdrDE electron transfer through Mph2 to result in movement of protons across the membrane via a quinol loop-like mechanism the cartoon needs to be modified so that the Mp accept electrons on the cytoplasmic face from Rnf and get oxidized by HdrDE on the periplasm (it is currently drawn the other way).

Reviewer #2

(Remarks to the Author)

Dear authors,

I enjoyed reading the manuscript that describes a syntrophic co-culture of a syntrophic acetate oxidizer and methanogen, originating the Baltic Sea sediments. The manuscript is advancing the field of microbe-mineral interactions and EET by describing the electron transport in genetic level in both partners, the electrogen (SAO) and methanogen. It describes a novel player, a MAG of *Candidatus Geosyntrophus acetoxidans*, its metabolic machinery for electron transfer to conductive particles (here, granular activated carbon) and the position of the novel candidate species in the phylogenomic tree. Similarly, the other member of the syntrophic team, novel type II methanogen *Methanosarcina* MAG, is described with its genomic blueprint for retrieving electrons from conductive particles and using them for methanogenesis. The manuscript also unravels several potential genomic markers that can be used to detect CIET in natural settings/in-situ. Overall, I found the study well-thought and analysis methodology accurate and meaningful. The manuscript is also well-written and clear. I only have some minor comments and notes that I hope the authors take into consideration:

l.27: ANI/AAI are abbreviated in the abstract, but not opened here but later on in the results and discussion section.

l.47-48: "SAO is essential in methane-producing ecosystems." I assume this statement builds upon the references mentioned in the previous sentence, but I still would like to see a reference for this too.

l. 160-162, 879-880 and 915-916: *Ca. Geosyntrophus acetoxidans* clearly delineates from the main branch of *Geobacter* according to both GTDB marker protein set (Fig.2b) and specific outer-surface EET conduit amino acid sequence (Fig.4b). Thus, in my opinion it could be named without the "Geo" to distinguish it from the other *Geobacteraceae*. Of course it can't be *Syntrophus* either but I don't know whether it would make sense to name it something like *Ca. (Pseudo)Pelosyntrophus*. That said, there is the other *Geobacter* (*psycrophilus*) together with it so I don't know whether it will be less or more confusing, and as I'm not an expert in naming novel candidate species I leave it up to the authors.

l. 880: is the "per site" here a typo?

l. 210-212: Could this have happened also during the over 9 year period of living in minimal media? Or is the time span still too short? Either case, it could be discussed a bit here.

l. 237-241: This struck me while I was reading the manuscript, so there seems to be an earlier paper describing this syntrophic relationship between this SAO and methanogen. Without reading it, I felt that this statement dampens down a bit the message of this manuscript. What I suggest is to add a bit more description of this earlier paper in the introduction part, and then highlighting the knowledge gap this research done here closes (e.g., MAGs and the genomic revelation of the cellular machineries related to CIET in both electrogen and methanogen).

l.274-278 Description of these environmentally available geoconductors could be added in the introduction (they are now very nicely described in the implications part, so if it's possible/there's no room for explanation in the intro, maybe add here a reference to that part, for example "see below in Implications").

l. 303 add "in" to "...may be involved IN organic matter.."

l. 333 and 334: 30-st transfer-> should this be 30th? 6-th ->6th as sixth transfer?

l. 369: should the sentence refer to all of the papers from 63-68? Or only the two? Just checking from the titles of the references...

l. 370-377: It would be good if these different studies/sites could be referenced.

l. 400: So assuming from these numbers, the transfers were done approximately 2 times a year. If so, it could be said here (transfers were done approximately twice a year/biannually/semiannually?)

l. 418: periodically-> same as in the Fig.1 (0, 14, 21, 28...)? You could add here a reference to the figure.

I. 448: how were the colors added to HIM images?

I. 559: OAT-> is this an abbreviation? If it is, could it be spelled out?

I. 574-575: SAMN47284751 (MAGs from GAC-incubations) but in NCBI description this is from the cultures which are inoculated with Magnetite nano powder? SAMN47284750 is for GAC? This should be corrected either in the manuscript or in the NCBI description, please check!

Figure 3. This is a very nice figure of the model on electron transfer from SAO to methanogen! For clarity, I suggest adding the electrons as is described in the text : 4 e- "balloons" in the centre and how these would be separated from MmcA (two for Rnf complex and two for MPH2 and so on), and maybe take the archaella pathway out from the figure, because to me it makes it a bit confusing. The text part describing the potential role of archaella is fine and understandable.

Overall, great job with a really long timespan culturing and gathering the (meta)genomic evidence for this exciting but still very much overlooked energy metabolism strategy.

Version 1:

Reviewer comments:

Reviewer #1

(Remarks to the Author)

The authors have addressed all of the major issues. A small point remains unaddressed due to a misunderstanding of the comment regarding the electron flow depicted in Fig 3. The comments:

"R1-10: If you want the Rnf->HdrDE electron transfer through Mph2 to result in movement of protons across the membrane via a quinol loop-like mechanism the cartoon needs to be modified so that the Mp accept electrons on the cytoplasmic face from Rnf and get oxidized by HdrDE on the periplasm (it is currently drawn the other way).

A1-10: We thank the reviewer for this observation. Our figure already shows the intended electron flow. MP is reduced by electrons from Rnf to form MPH₂ within the cytoplasmic membrane, as indicated by the arrows from MP to MPH₂. Methanophenazine is a membrane-bound carrier, similar to menaquinone in bacterial membranes, so it is drawn inside the membrane rather than on the cytoplasmic side and diffuses laterally to HdrE, which is also located in the cytoplasmic membrane. The red arrow from Rnf shows electrons being provided to this MP pool, and the reduced MPH₂ then donates electrons to HdrE. This depiction follows published models of DIET/EET in *Methanosarcina acetivorans*, so the figure was already correct and has not been modified."

The comment was referring to WHICH face of the membrane the oxidation and reduction occur on. You are claiming that "membrane HdrDE reduces CoM-S-S-CoB via the methanophenazine pool with scalar H⁺ release to the periplasm-like space, rebuilding the ion motive force" line 299.

The current version of Fig 3. illustrates the loops of Mph2 and Mp at an angle, and the reason that one would draw these loops at an angle is to indicate WHICH side of the membrane you think the oxidation and reduction are occurring. Currently that loop is oriented in the wrong direction to effectuate proton release into the periplasmic space. The reduction is drawn at Rnf on the periplasmic side of the membrane, while the oxidation is drawn on the cytoplasmic side. That would result in a net loss of pmf, because you would be taking up a pair of protons from the periplasm at Rnf and releasing that into the cytoplasm at Hdr. If you want proton release to the periplasm should draw Mph2 being oxidized on the periplasmic side, it is a b-type cytochrome after all, it can facilitate the electrons passing from the periplasmic side through the membrane to the hdrD for heterodisulfide reduction. Review: <https://doi.org/10.1016/j.bbabi.2008.09.008> for the logic of quinol loops. HdrDE is NOT a proton pump, although it is currently drawn as if a proton is moving along a big bold arrow through the complex. Fpo and Rnf should have those big bold arrows because they are actually ion pumps. Its a minor point, one doesn't necessarily need to go into this much detail in a cell diagram, but if you do choose to have the detail of a loop drawn at an angle, it should be pointing in the right direction.

Reviewer #2

(Remarks to the Author)

I'm satisfied with the modifications and explanations the authors have given to the reviewers' comments and thank the authors for taking into consideration the suggestions made. On my behalf, the revised manuscript is of high quality and I'd be happy to support its publication.

Version 2:

Reviewer comments:

Reviewer #1

(Remarks to the Author)

Great, fig updated as suggested. Good for publication.

Reviewer comments

Reviewer #1 (Comments for the Author):

R1-1: Jovicic and Anestis et al report on the metagenomic characterization of a methanogenic enrichment culture from the Baltic Sea. Prior work has demonstrated the importance of granular activated carbon for high methane production rates in these incubations that arises from a syntrophic relationship between one or two main syntrophic acetate oxidizing bacteria and a single electron accepting methanogen. With the addition of the metagenomic information provided here, Jovicic and Anestis et al provide a hypothesis for how this symbiosis is carried out and compare it with cultivated relatives of the symbionts.

This manuscript adds detail to an interesting system, the methods are sound, and it is good to have genomic follow-up to prior work on these organisms (i.e. ref 15 and 46). There are a few issues that should be addressed before the work is suitable for publication, outlined below.

A1-1: We thank the reviewer for the positive and encouraging feedback.

R1-2: A major promise made up front in this manuscript is that “This first genomic blueprint of a natural CIET-SAO consortium identifies potential genomic markers (distinct PCCs, MmcA) for in-situ detection..” (Abstract) and later: “The molecular signatures identified here (novel PCC-clusters and MmcA variants) provide some of the first genomic markers for targeted detection.” (Implications).

It would be fantastic to have some sort of a diagnostic gene(s) to positively identify the CIET process in genomes/metagenomes. Unfortunately, that claim is delivered on in this manuscript. For a gene to be a diagnostic marker that can be used for targeted detection it needs to either A) be completely unique to that metabolism with no homologs in other organisms, or B) if there are homologs/paralogs outside of organisms that carry out that metabolism, you need to have clearly defined some specific phylogenetic subgroup(s) that correspond to your process of interest. Then, if a novel sequence is found to fall within those groups, you can propose that they too might serve that function.

A) is clearly not the case, both PCCs and MmcA are used by organisms that do not rely on CIET. Yet the authors have not at all attempted to define the groups for option B), so this work in its present form has not delivered on this major claim. While Figures 4 and 5 have phylogenetic trees with colored subclades, this does not help...

Nothing presented actually delivers on a useful test for CIET in a genome, so unless the authors come up with a clear explanation supporting this claim it needs to be removed from

the manuscript. The methanosarcina looks just like a normal methanosarcina in every way described in the manuscript, and the Geosyntrophus maybe has fewer PCCs than some Geobacter species, but there is no way to look at a genome, see that it has a single PCC, and tell what it is specific for.

A1-2: We appreciate the reviewer's concerns. We would like to clarify that the manuscript does not claim to have identified diagnostic gene markers for CIET. Throughout the manuscript, we have used terms such as "*potential*" or "*candidate*" genomic signatures, with one earlier inconsistency in the Implication section, which has now been corrected.

Our intention was to highlight why certain genome features (PCC clusters and *mmcA* variants) are genomically noteworthy, not to claim that they are sufficient for diagnostic use. Their relevance rests on two empirical observations:

- (i) They occur consistently during CIET- but are absent from non-CIET enrichments from the same environments. (this work and additional work by Anestis et al., Manuscript 1 *In Preparation*)
- (ii) PPCs and *mmcA* show distinct expression patterns under CIET vs. non-CIET conditions in both synthetic and environmental consortia (Anestis et al., Manuscript 1&2 *In Preparation*).

These transcriptomic data (part of a broader, ongoing effort) fall outside the scope of the present genome-centric study but demonstrate that these genomic features have functional relevance to CIET metabolism, guiding future development of potential diagnostic criteria.

These genomic features (PCCs, *mmcA*) therefore help flag putative CIET capable taxa in metagenomes. Nevertheless, in this Baltic consortium CIET has been already experimentally confirmed (Rotaru et al., mBio 2018, 9(3), 10-1128).

We have revised the Abstract, Results, and Discussion to refer to these uniformly as "*genomic features*" avoiding any implication that these constitute validated diagnostic markers.

R1-3: For *MmcA* the Baltic one is very closely related to those in methanosarcina that grow in pure culture using a variety of methanogenic pathways, so there's nothing about *MmcA* that can be called diagnostic for CIET.

A1-3: We thank the reviewer for this comment. We never referred to *mmcA* as a diagnostic gene for CIET. We note that *MmcA* has been shown to act as a bidirectional electron transfer conduit in type II *Methanosarcina* (*M. acetivorans*). The *mmcA*-gene is upregulated significantly during EET-growth (4-5-fold with Fe⁰-electrons or DIET partners), in contrast to

soluble electron donors (acetate, methanol). Deletion of *mmcA* abolished EET, yet it did **NOT** impair growth on soluble substrates, directly confirming its specific role in EET.

Thus, while the *mmcA* gene alone cannot be considered a diagnostic gene for CIET (and we did not claim it to be), its presence indicates functional EET/CIET potential, especially when interpreted together with complementary features of the syntrophic partners such as PCCs and pili.

We have added a paragraph clarifying this point (lines 265-277).

R1-4: For the *omcB*, it is claimed to be quite different than ones in *G. sulfurreducens* in terms of amino acid identity, but that is fairly standard for multiheme cytochromes, they evolve incredibly fast. Are we to take the *omcB* subgroup that is colored blue in Fig 4 as the “CIET group”? Do we know that the *Oryzomonas* sp. therein use them for CIET specifically? Doubtful, and the authors have not made that claim.

A1-4: We sincerely thank the reviewer for this insightful comment. It prompted us to re-examine the PCC architecture and *omcB*'s genomic context in greater detail, including synteny, evolutionary origin, and evolutionary rates. Through these additional analyses, we identified a second PCC cluster in *Ca. Geosyntrophus*, whereas its closest relative *G. psychrophilus* contains only one PCC module lacking the *ombB*–*omaB*–*omcB* set.

These new findings have now been incorporated into the revised manuscript (lines 197-211).

Regarding the phylogeny in Fig. 4, we agree that MHCs diversify rapidly and that amino-acid identity is often low. However, we show that despite the low aminoacid identities, *Ca. Geosyntrophus* retains two PCC-clusters with comparable synteny and heme-binding site content to those required for EET in *Geobacter sulfurreducens*. The outer-membrane cytochrome in one PCC has even expanded (doubled) its heme content, with the extracellular MHCs displaying the highest evolutionary rates, consistent with rapid evolution of extracellular proteins exposed to environmental selection pressure.

These new findings were included in Fig. 4, Supplementary Note and the revised manuscript text (lines 213-219).

We did not interpret the blue-shaded *OmcB* group in the tree as a “CIET clade,” nor do we imply that the *Oryzomonas* sequences within the subtree participate in CIET. Their placement simply reflects evolutionary relatedness among *Desulfuromonadales* *OmcB* homologues. Besides, *Oryzomonas* is not the closest relative of *Ca. Geosyntrophus*. Its EET capabilities are not characterized. *Oryzomonas* genomes lack the porin component of the PCC-

architecture, meaning their clusters are not syntenic with those of *Geobacter sulfurreducens* or *Ca. Geosyntrophus*. We decided against adding additional discussions about *Oryzomonas* and remain focused on *G. psychrophilus* (as closest relative) and the model organism *G. sulfurreducens*, where the PCC-clusters are best understood.

Reviewer #1 (Minor comments)

R1-5: Line 169-170 “neither of the constituents showed significant sequence identity to any canonical *Geobacter*.” What does “significant” mean, how are they on a tree with *Geobacter* versions in Fig 4b if there is no significant sequence similarity?

A1-5: We appreciate the reviewer’s request for clarification. We removed the statement “no significant sequence identity”. Our intention was to explain that the extracellular conduit showed low amino acid sequence homology to *Geobacter* MHCs.

As shown now in Fig. S10 (below), many extracellular MHCs fall in the 30–40% aa identity range, consistent with low but detectable homology and fast evolutionary rates. This level of amino acid identity is sufficient for reliable phylogenetic inference.

Figure S10 (Supplementary Note)

This has now been made clearer in the manuscript text (lines 197-211). Additional information can be found in Supplementary Data Tables S1-S2 and Supplementary Note Fig. S9.

R1-6: Line 189-190 “indicating independent evolution of an identical solution.” It is very unclear what is being proposed here from an evolutionary standpoint. Is the claim that the gene organization shown in Fig 4a is a completely novel invention of the PCC complex? This seems like a very strange conclusion with such clear gene synteny and homology. This is almost

certainly a mixture of rapid evolution of paralogs and/or HGT not a completely de novo “independent” evolution.

A1-6: We thank the reviewer for this comment. We agree that the wording “independent evolution of an identical solution” was unclear. Our intention was to emphasize that the PCC-clusters have diverged substantially at the sequence level from those of canonical *Geobacter* PCCs while retaining conserved gene synteny and overall functional architecture.

We have revised the text (lines 213-219) to reflect this more accurate interpretation.

R1-7: Line 346-347 “PCC clusters that are unrelated to those of canonical *Geobacter* pointing at a convergent evolution of EET machineries”. Sorry to keep harping on this, but it is extremely odd to talk about things being “unrelated” when you have pulled them out with Blast, aligned them to one another and put them on a phylogenetic tree together. This is the rapid evolution of cytochrome c paralogs which is just how these things go, no clear explanation here of how this is “convergent”.

A1-7: We thank the reviewer for this correction. We fully agree that our original wording was incorrect. The reviewer’s comments prompted us to re-examine all MHCs and PCC clusters in *Ca. Geosyntrophus* and reassess their evolutionary relationships with *Geobacter* homologues.

We have now added % aa identities for all MHCs in the Supplementary Data Table S2.

These analyses make it clear that “convergent evolution” and “unrelated” were inaccurate terms. With only a few exceptions, the 47 MHCs of *Ca. Geosyntrophus* show clear homology to *Geobacter* MHCs, some syntenic clusters pointing to orthologous relationships, others representing more diverged homologues, consistent with divergent evolution of shared ancestral EET modules rather than convergence.

We have revised the text accordingly to reflect this more accurate interpretation (lines 213-219).

R1-8: Line 243: “we recovered the MAG of this methanogen...” already been discussing this MAG for a few paragraphs, introducing it here doesn’t make sense.

A1-8: Changed as suggested.

R1-9: Line 204: “disposed of the resulting electrons through a streamline, particle-obligate EET-conduit.” Has a control been done to show that the *Geosyntrophus* cannot put electrons onto iron oxides or other electron acceptors? The authors cite prior work that shows the *Methanosarcina* cannot grow directly on acetate even though it has the machinery for it, so there is some experimental support for the claim that the *methanosarcina* is specialized for CIET. But it seems like a similar experiment would need to be done, but for the bacteria and alternative electron acceptors, before you can claim that *Geosyntrophus* is specialized for conductive particles over other electron acceptors.

A1-9: We thank the reviewer for this comment. The phrase that triggered the concern (“*streamlined, particle-obligate EET conduit*”) was an unintended wording issue, and we have removed “*particle-obligate*” from the revised manuscript to avoid implying exclusivity of the conduit toward conductive particles.

Our intended point was different: we compared the reduced EET-conduit repertoire of *Ca. Geosyntrophus* to the highly redundant systems typical of canonical *Geobacter* species, and we stated only that this reduced architecture *may suggest* niche specialization. We did **not** conclude in our manuscript that conductive particles are the *only* electron acceptor the organism can use. What we know is that *Ca. Geosyntrophus* uses conductive particles during CIET but cannot switch to DIET in the absence of conductive particles.

Due to this misunderstanding, the reviewer pointed at a potential control looking at alternative electron acceptors for a *Ca. Geosyntrophus* that has not been yet isolated. Testing of alternative electron acceptors is outside the scope of this genome-centric study looking at a conductive particle mediated interspecies interaction.

We made sure NOT to state that the EET-conduit is conductive particle specific in the manuscript text.

R1-10: If you want the Rnf->HdrDE electron transfer through MpH₂ to result in movement of protons across the membrane via a quinol loop-like mechanism the cartoon needs to be modified so that the Mp accept electrons on the cytoplasmic face from Rnf and get oxidized by HdrDE on the periplasm (it is currently drawn the other way).

A1-10: We thank the reviewer for this observation. Our figure already shows the intended electron flow. MP is reduced by electrons from Rnf to form MPH₂ within the cytoplasmic membrane, as indicated by the arrows from MP to MPH₂. Methanophenazine is a membrane-bound carrier, similar to menaquinone in bacterial membranes, so it is drawn inside the membrane rather than on the cytoplasmic side and diffuses laterally to HdrE, which is also located in the cytoplasmic membrane. The red arrow from Rnf shows electrons being provided to this MP pool, and the reduced MPH₂ then donates electrons to HdrE. This depiction follows published models of DIET/EET in *Methanosarcina acetivorans*, so the figure was already correct and has not been modified.

General:

We thank Reviewer 1 for their thoughtful evaluation. The comments on the evolutionary relationships were particularly helpful and prompted us to carry out additional bioinformatic analyses that now support the statements in the manuscript, thereby strengthening it.

Continued next page with replies to reviewer 2

Reviewer #2 (Remarks to the Author)

R2-1: Dear authors, I enjoyed reading the manuscript that describes a syntrophic co-culture of a syntrophic acetate oxidizer and methanogen, originating the Baltic Sea sediments. The manuscript is advancing the field of microbe-mineral interactions and EET by describing the electron transport in genetic level in both partners, the electrogen (SAO) and methanogen. It describes a novel player, a MAG of *Candidatus Geosyntrophus acetoxidans*, its metabolic machinery for electron transfer to conductive particles (here, granular activated carbon) and the position of the novel candidate species in the phylogenomic tree. Similarly, the other member of the syntrophic team, novel type II methanogen *Methanosarcina* MAG, is described with its genomic blueprint for retrieving electrons from conductive particles and using them for methanogenesis. The manuscript also unravels several potential genomic markers that can be used to detect CIET in natural settings/in-situ. Overall, I found the study well-thought and analysis methodology accurate and meaningful. The manuscript is also well-written and clear. I only have some minor comments and notes that I hope the authors take into consideration:

A2-1: We thank the reviewer for the positive and encouraging feedback.

R2-2: I.27: ANI/AAI are abbreviated in the abstract, but not opened here but later on in the results and discussion section.

A2-2: Thank you for catching this. We have revised the abstract and completely removed the ANI/AAI for space considerations.

R2-3: I.47-48: "SAO is essential in methane-producing ecosystems." I assume this statement builds upon the references mentioned in the previous sentence, but I still would like to see a reference for this too.

A2-3: We have added references to this statement as advised.

R2-4: I. 160-162, 879-880 and 915-916: *Ca. Geosyntrophus acetoxidans* clearly delineates from the main branch of *Geobacter* according to both GTDB marker protein set (Fig.2b) and specific outer-surface EET conduit amino acid sequence (Fig.4b). Thus, in my opinion it could be named without the "Geo" to distinguish it from the other *Geobacteraceae*. Of course it can't be *Syntrophus* either but I don't know whether it would make sense to name it something like

Ca. (Pseudo)Pelosyntrophus. That said, there is the other *Geobacter* (psychrophilus) together with it so I don't know whether it will be less or more confusing, and as I'm not an expert in naming novel candidate species I leave it up to the authors.

A2-4: We appreciate the reviewer's thoughtful reflections regarding the genus name. As part of a broader comparative analysis across all sequenced members of the class *Desulfuromonadia* (Anestis et al., submitted), we found that the taxonomy of this entire class requires substantial revision, including clarification of multiple genera. The family name "Pseudopelobacteraceae" is not widely accepted, and no validly named organism "Pseudopelobacter" exists, making genus-level linkage to that name problematic.

We therefore chose the name '**Candidatus Geosyntrophus acetoxidans**' based on etymological reasoning rather than family associations. The root **geo-** derives from the Greek *gē*, meaning *earth/ground/subsurface*, and is widely used across microbiology for organisms originating from soils, sediments or subsurface. Its use is not restricted to *Geobacter* or to the family *Geobacteraceae*. Examples include *Geoglobus* (Archaeoglobi), *Geofilum* (Bacteroidia), and *Geopsychrobacter* (*Desulfuromonadia*), all of which originate from sedimentary, benthic, or subsurface environments.

The second element, 'syntrophus', reflects the organism's obligate syntrophic lifestyle, and the species epithet 'acetoxidans' denotes its role as an acetate oxidizer. Thus, the name describes the organism clearly and accurately as a sediment-associated syntroph that oxidizes acetate. To ensure this is immediately clear to readers, we have added the etymology of the *Candidatus* name in the manuscript (lines 169-172): "(Candidatus Ge.o.syn.tro'phus; Gr. n. *gē*, earth; Gr. adj. *syntrophos*, feeding together; *acet.oxidans*: N.L. n. *acetum*, acetate, and N.L. pres. part. *oxidans*, oxidizing)"

R2-5: l. 880: is the "per site" here a typo?

A2-5: Yes, thank you for noting this. "Per site" was a typo and has been removed in the revised version.

R2-6: l. 210-212: Could this have happened also during the over 9 year period of living in minimal media? Or is the time span still too short? Either case, it could be discussed a bit here.

A2-6: We thank the reviewer for this interesting comment. While adaptive streamlining can occur during long-term laboratory cultivation in fast-growing microorganisms like *E. coli*, a

nine-year period is unlikely to be sufficient for the major genomic rearrangements and reduction in the *Ca. Geosyntrophus* within this slow-growing consortium. As suggested by reviewer 1, we re-examined the evolutionary origins of all MHCs in the *Ca. Geosyntrophus* genome, which led us to identify a second PCC cluster. Even with two PCC clusters, however, this organism still encodes only half the PCCs found in canonical *Geobacter* species, and roughly half the number of MHCs for the same size genome (~3.8Mb). *Geosyntrophus*' reduced EET repertoire is consistent with long-term genome streamlining in its original sediment niche and is unreasonable to have arisen during the 9–10 years of laboratory incubation.

We have added information to the text regarding the second PCC cluster and the reduced MHC repertoire (lines 197-219). Although we acknowledge the reviewer's comment as a valid consideration for fast-growing communities, for this slow-growing consortium we chose not to discuss it in the manuscript.

R2-7: l. 237-241: This struck me while I was reading the manuscript, so there seems to be an earlier paper describing this syntrophic relationship between this SAO and methanogen. Without reading it, I felt that this statement dampens down a bit the message of this manuscript. What I suggest is to add a bit more description of this earlier paper in the introduction part, and then highlighting the knowledge gap this research done here closes (e.g., MAGs and the genomic revelation of the cellular machineries related to CIET in both electrogen and methanogen).

A2-7: We thank the reviewer for this observation. Our earlier study (Rotaru et al., 2018; ref. 15) was described in detail in the Introduction and through the manuscript, where we summarized the physiological evidence for the CIET-SAO partnership and explicitly outlined the knowledge gap addressed in the present work. In response to the reviewer's suggestions, we have tried to explain better the differences between the previous work and the present work. For this purpose, we have added a brief clarifying sentence in the Introduction (lines 88-91), one at the start of the Results section (lines 102-103), and lastly, we reformulated the specific lines that prompted the comment (lines 279-289). We hope these additions make the links and differences between the two studies clearer.

R2-8: l.274-278 Description of these environmentally available geoconductors could be added in the introduction (they are now very nicely described in the implications part, so if it's

possible/there's no room for explanation in the intro, maybe add here a reference to that part, for example "see below in Implications").

A2-8: We thank the reviewer for this suggestion. A description of environmentally available geoconductors (e.g., magnetite and wildfire-derived char particles) is already included in the Introduction, where we briefly introduce these materials in the context of coastal sediment processes. We carefully checked the introduction to ensure the description is clear. No further changes were necessary.

R2-9: l. 303 add "in" to "...may be involved IN organic matter.."

A2-9: Changed as suggested.

R2-10: l. 333 and 334: 30-st transfer-> should this be 30th? 6-th ->6th as sixth transfer?

A2-10: Changed as suggested.

R2-11: l. 369: should the sentence refer to all of the papers from 63-68? Or only the two? Just checking from the titles of the references...

A2-11: We thank the reviewer for noticing this. The two references were specific to wildfires, fossil fuel combustion and industry as sources that rise black carbon (BC) in the environment. However, because the way the references were listed created confusion, we moved them at the end of the sentence, so we could include those two references alongside references describing BC distribution in coastal, glacier and riverine environments.

R2-12: l. 370-377: It would be good if these different studies/sites could be referenced.

A2-12: References were included at the appropriate locations as advised.

R2-13: l. 400: So assuming from these numbers, the transfers were done approximately 2 times a year. If so, it could be said here (transfers were done approximately twice a year/biannually/semiannually?)

A2-13: Changed as suggested.

R2-14: l. 418: periodically-> same as in the Fig.1 (0, 14, 21, 28...)? You could add here a reference to the figure.

A2-14: Changed as suggested.

R2-15: l. 448: how were the colors added to HIM images?

A2-15: We thank the reviewer for this question. The HIM images were colorized manually in BioRender; this information has now been added to the Materials and Methods (lines 490-491). The original, unmodified HIM images are provided in the supplementary material. We applied colorization to distinguish cell types more clearly, as users with visual impairments reported difficulty differentiating gray cells against the gray carbon background.

R2-16: l. 559: OAT-> is this an abbreviation? If it is, could it be spelled out?

A2-16: Thank you for noticing. We forgot to introduce the abbreviation, OrthoANI Tool (OAT).

R2: l. 574-575: SAMN47284751(MAGs from GAC-incubations) but in NCBI description this is from the cultures which are inoculated with Magnetite nano powder? SAMN47284750 is for GAC? This should be corrected either in the manuscript or in the NCBI description, please check!

A: We thank the reviewer for catching this mix-up. This was an error in the manuscript only. It has now been corrected so that SAMN47284750 refers to the GAC incubation, consistent with the NCBI metadata.

R2: Figure 3. This is a very nice figure of the model on electron transfer from SAO to methanogen! For clarity, I suggest adding the electrons as is described in the text : 4 e⁻ "balloons" in the centre and how these would be separated from MmcA (two for Rnf complex and two for MPH2 and so on), **and maybe take the archellum pathway out from the figure, because to me it makes it a bit confusing.** The text part describing the potential role of archaella is fine and understandable.

A: We thank the reviewer for the thoughtful suggestions and are very glad the figure is appreciated. At this stage, we prefer not to modify it because we believe it is already too complex. Besides, drawing the 4e⁻ split would imply a mechanistic resolution that is only hypothesized and prefer not showing it in a conceptual electron flow model. We chose to retain

the archaeella because co-culture studies with archaeella-deletion mutants have shown that this structure is crucial for DIET interactions in type 2 Methanosarcina, even though its conductivity and exact contribution to electron flow remain unknown. We hope the reviewer agrees that maintaining the current level of abstraction preserves clarity without overinterpreting the available data.

Overall, great job with a really long timespan culturing and gathering the (meta)genomic evidence for this exciting but still very much overlooked energy metabolism strategy.

We appreciate the reviewer's thoughtful comments and encouragement. The feedback has been very helpful, and we believe the manuscript has improved substantially as a result.

8 January 2026

REPLY TO REVIEWER COMMENTS

Reviewer #1 (Remarks to the Author):

The authors have addressed all of the major issues. A small point remains unaddressed due to a misunderstanding of the comment regarding the electron flow depicted in Fig 3. The comments:

"R1-10: If you want the Rnf->HdrDE electron transfer through MpH₂ to result in movement of protons across the membrane via a quinol loop-like mechanism the cartoon needs to be modified so that the Mp accept electrons on the cytoplasmic face from Rnf and get oxidized by HdrDE on the periplasm (it is currently drawn the other way).

A1-10: We thank the reviewer for this observation. Our figure already shows the intended electron flow. MP is reduced by electrons from Rnf to form MPH₂ within the cytoplasmic membrane, as indicated by the arrows from MP to MPH₂. Methanophenazine is a membrane-bound carrier, similar to menaquinone in bacterial membranes, so it is drawn inside the membrane rather than on the cytoplasmic side and diffuses laterally to HdrE, which is also located in the cytoplasmic membrane. The red arrow from Rnf shows electrons being provided to this MP pool, and the reduced MPH₂ then donates electrons to HdrE. This depiction follows published models of DIET/EET in *Methanosarcina acetivorans*, so the figure was already correct and has not been modified."

The comment was referring to WHICH face of the membrane the oxidation and reduction occur on. You are claiming that "membrane HdrDE reduces CoM-S-S-CoB via the methanophenazine pool with scalar H⁺ release to the periplasm-like space, rebuilding the ion motive force" line 299.

The current version of Fig 3. illustrates the loops of MpH₂ and Mp at an angle, and the reason that one would draw these loops at an angle is to indicate WHICH side of the membrane you think the oxidation and reduction are occurring. Currently that loop is oriented in the wrong direction to effectuate proton release into the periplasmic space. The reduction is drawn at Rnf on the periplasmic side of the membrane, while the oxidation is drawn on the cytoplasmic side. That would result in a net loss of pmf, because you would be taking up a pair of protons from the periplasm at Rnf and releasing that into the cytoplasm at Hdr. If you want proton release to the periplasm should draw MpH₂ being oxidized on the periplasmic side, it is a b-type cytochrome after all, it can facilitate the electrons passing from the periplasmic side through the membrane to the hdrD for heterodisulfide reduction. Review: <https://doi.org/10.1016/j.bbabi.2008.09.008> for the logic of quinol loops. HdrDE is NOT a proton pump, although it is currently drawn as if a proton is moving along a big bold arrow through the complex. Fpo and Rnf should have those big bold arrows because they are actually ion pumps. Its a minor point, one doesn't necessarily need to go into this much detail in a cell diagram, but if you do choose to have the detail of a loop drawn at an angle, it should be pointing in the right direction.

Answer to comment by reviewer 1:

We thank reviewer 1 for the careful read of our revised work. Our previous reply to comment 10 posed by reviewer 1 was indeed a misunderstanding as we focused on methanophenazine being membrane-bound, but we did not address the key point about which face of the membrane MP is reduced/oxidized. We have therefore revised Fig. 3 to make the sidedness explicit and consistent with the methanophenazine/HdrDE redox-loop (scalar H⁺) model described for Type II *Methanosarcina* (e.g., *M. acetivorans*). Specifically, MP is now shown to be reduced close the cytoplasmic facing-side of the Rnf, and MpH₂ is shown to be oxidized on the periplasm-like facing-side of HdrDE with scalar H⁺ release to the periplasm-like space. We removed the thick H⁺ arrow through HdrDE to avoid implying that HdrDE is an ion pump, and we adjusted the MP/MpH₂ loop orientation and labels accordingly. The manuscript text already refers to scalar H⁺ release, and the revised figure now matches this description.

Reviewer #2 (Remarks to the Author):

I'm satisfied with the modifications and explanations the authors have given to the reviewers' comments and thank the authors for taking into consideration the suggestions made. On my behalf, the revised manuscript is of high quality and I'd be happy to support its publication.

Answer to comment by reviewer 2: We thank reviewer 2 for taking the time to review our revised work.